# The Gut–Brain–Microbiome Axis in Bumble Bees

**DOI:** 10.3390/insects11080517

**Published:** 2020-08-10

**Authors:** Laura Leger, Quinn S. McFrederick

**Affiliations:** Department of Entomology, University of California, Riverside, CA 92521, USA; llege001@ucr.edu

**Keywords:** insect cognition, free moving proboscis extension response (FMPER) assays, germ free bees

## Abstract

The brain-gut–microbiome axis is an emerging area of study, particularly in vertebrate systems. Existing evidence suggests that gut microbes can influence basic physiological functions and that perturbations to the gut microbiome can have deleterious effects on cognition and lead to neurodevelopmental disorders. While this relationship has been extensively studied in vertebrate systems, little is known about this relationship in insects. We hypothesized that because of its importance in bee health, the gut microbiota influences learning and memory in adult bumble bees. As an initial test of whether there is a brain-gut–microbiome axis in bumble bees, we reared microbe-inoculated and microbe-depleted bees from commercial *Bombus impatiens* colonies. We then conditioned experimental bees to associate a sucrose reward with a color and tested their ability to learn and remember the rewarding color. We found no difference between microbe-inoculated and microbe-depleted bumble bees in performance during the behavioral assay. While these results suggest that the brain-gut–microbiome axis is not evident in *Bombus impatiens*, future studies with different invertebrate systems are needed to further investigate this phenomenon.

## 1. Introduction

The relationship between the gut and brain, also known as the gut–brain axis, is an emerging area of study, especially in vertebrate systems. In vertebrates, the gut has been shown to communicate with the central nervous system via immune, endocrine, and neural pathways [1]. This gut–brain axis influences basic physiological functions like satiety, metabolism, and the regulation of host homeostasis [2]. Recent research has shown that gut microbes can contribute to this gut–brain axis and that perturbations to the microbial community can have detrimental effects on learning, memory, and neurological disorders in vertebrate hosts [3,4]. Although these relationships have received increasing attention in vertebrate systems, understanding the mechanisms for them remains difficult due to the large diversity of symbionts that vertebrates can host [5]. Because they host a more tractable number of microbial symbionts [6], insects have been suggested as an ideal system to study the brain-gut-microbiome axis [7].

The fact that microorganisms can alter behavioral phenotypes in insects is not a novel concept. For example, *Wolbachia*, a common insect endosymbiont, alters mating preference in infected hosts [8] and pathogenic fungi like *Ophiocordyceps* alter insect behavior in order to promote their own transmission [9]. Gut microbial communities can contribute to nestmate recognition in some termites and ants [10,11]. While there is evidence that microorganisms that can alter host behavior, few studies have explored the brain-gut–microbiome axis in insect systems. A recent study, however, showed that learning and memory in *D. melanogaster* can be disrupted by knocking down the gut microbiota and can be rescued with as few as two bacterial species [12]. The mechanism and function of a gut–brain axis may, therefore, be evolutionarily conserved across vertebrates and insects. Another study of *D. melanogaster* suggests that gut microbes can influence mating preference by altering the production of cuticular hydrocarbons when flies are raised on diets with different microbial composition [13]. As the gut microbiota plays an important role in many invertebrate species, exploring the gut–brain axis in other insect systems warrants further investigation.

Learning and memory play an important role in both social and solitary bee species [14]. Bees depend on their ability to forage efficiently, locating plentiful resources and returning to their nest sites, to feed themselves and their developing brood [15]. Honey bees have been suggested to be ideal hosts to test the generality of the brain-gut-microbiome axis across animals because there is mounting evidence that insect behavior may be influenced by mechanisms that are, in part, homologous to those in mammals [7]. As bumble bees share ancestry and a related core gut microbiome with honey bees [16], we extend Liberti and Engel’s [7] recommendation to the corbiculate bees, which include bumble bees and honey bees. The corbiculate apids host a small number of gut symbionts that carry similar functions to those of vertebrates [17]. For example, the gut microbial community in honey bees has been shown to contribute to gustatory response, increase weight gain, and process carbohydrates into short chain fatty acids [18]. Additionally, honey bee gut symbionts have been shown to metabolize toxic sugars that can be found in nectar [19]. The gut microbiota in bumble bees protects against the gut parasite *Crithidia bombi* [20]. The gut microbiome in bees is relatively easy to manipulate; as bees undergo metamorphosis, they shed their larval gut lining and grow a new gut [6]. Newly emerged adult bees void the remnants of their larval gut and the bacteria that resided as meconium [21]. In our previous research, we found that withholding microbe inoculations from newly emerged bees that we kept in sterile conditions results in a greatly reduced gut microbiota without the use of antibiotics [22], making the bumble bee microbiome particularly well-suited for behavioral assays.

Here, we tested the hypothesis that because of its importance in corbiculate bee health [23,24,25], the gut microbiome can influence learning and memory in bumble bees. We reared microbe-depleted and microbe-inoculated bees from commercial *Bombus impatiens* colonies and subjected them to a visual learning assay to determine whether the absence of a gut microbiota could induce learning deficits.

## 2. Methods

### 2.1. Bumble Bee Rearing and Microbe-Inoculation Treatments

We performed two separate experiments, each with a different set of *Bombus impatiens* colonies from the same commercial provider, Koppert Biological Systems (Howell, MI, USA). The first experiment (round 1) resulted in a small sample size (*n* = 25 microbe-depleted bees, *n* = 26 microbe-inoculated bees). We therefore used the results of this first experiment to calculate an estimated effect size and conduct a power calculation which we used to guide the sample size of our second experiment. In the second experiment, we used the same methods with a new group of colonies at a different date. For the second experiment (round 2), we increased our sample sizes to *n* = 41 microbe-depleted bees and *n* = 39 microbe-inoculated bees. We kept all colonies under constant darkness at room temperature (~25 °C). Upon arrival, we selected five workers at random from each colony and screened these workers for the common bumble bee gut pathogen *Crithidia bombi* using microscopy. *Crithidia bombi* is a pathogen that influences learning when infection is intense in bumble bees [26]. We fed all colonies an approximately 3 gm pollen ball made with pollen mixed with 40% sucrose that we replenished every other day. We also provided the colonies with a 1.5 L bag of 40% sucrose solution (*w/v*) that was cleaned and replaced once per week.

In our previous work [22], we showed that our experimental methods do not produce truly gnotobiotic organisms but greatly reduce the abundance and diversity of microorganisms in the gut. To obtain experimental bees, we removed dark cocoons from colonies and aseptically removed the mature pupae from their cocoons [22]. We placed the pupae into sterile 48-well tissue plates until they emerged as adults under constant darkness and ~30% relative humidity at 27 °C. On the day the new adult bees emerged (day 0), we organized the bees into cohorts of 3–10 by colony of origin and date of emergence, and housed each cohort in separate, UV-sterilized 475 mL polypropylene containers (WebRestaurantStore, Lancaster, PA, USA).

On day 0, we prepared a microbe inoculum by harvesting the guts of 5 live bees from the colonies of origin. To feed newly emerged bees, we prepared a sterile sucrose pollen solution by mixing 1 gm of pollen per 100 mL of 40% sucrose which we then autoclaved. We placed the dissected guts into 500 µL of the sucrose pollen solution, which we then homogenized using sterile plastic pestles. For each cohort of microbe-inoculated bees, we spiked two 1.5 mL feeder tubes of the sucrose pollen solution with half of the 500 µL of gut homogenate. To ensure that bees acquired microbes from the inoculum, we allowed the bees to feed on the gut homogenate and sucrose pollen mixture ad libitum until they had consumed the entire amount or 48 h had passed (microbe-inoculated bees). When the solution was depleted, we replaced it with a fresh, sterile sucrose pollen solution. Generally, the groups of bees would finish the 3 mL of the microbe-spiked solution within 24 h. To promote the establishment of microbes in the gut, we then fed the bees with a sterile sucrose pollen solution until day 5 after emergence—the time reported for the gut microbiota to stabilize [27,28]. We used germ-free protocols that have previously been demonstrated to decrease the abundance and diversity of the bee gut microbiome by 10^4^ times [22]. Briefly, we fed the other half of the bees with the same sterile sucrose pollen solution without added gut homogenate (microbe-depleted bees). To control for age, microbe-depleted bees also fed ad libitum on the sterile sucrose pollen solution for 5 days. On day 5, we began feeding both treatments a sterile 40% sucrose solution without pollen for an additional 8-day experimental period.

### 2.2. Free Moving Proboscis Extension Response (FMPER) Assays

At the end of the experimental period, we performed a free-moving proboscis extension response (FMPER) assay to assess learning and memory in the experimental bees using established protocols [29]. Typically, traditional PER assays are used to test memory and learning in bees by constraining bees in harnesses and conditioning them to associate an odor with a sucrose stimulus [30,31]. We used the FMPER assay, which allows the bees to move freely in a less confined space, in order to reduce the stress of a lab-based behavioral assay. For this behavioral assay, we designed assay tubes using 15 × 2.5 cm clear acrylic tubes (S&W Plastics, Riverside, CA, USA). The tubes are open on one end and sealed on the other side with a clear piece of plastic containing two small puncture holes approximately 3 mm in diameter (Figure 1A,B). We recorded treatment and colony of origin for later analysis. In order to perform a blind assay, we randomly assigned individual bees to tubes with novel labels, noting only the number on the treatment cup which was concealed during the assay. We starved the bees in their assay tubes for 4–6 h before performing the assays.

Using classical conditioning, we trained the bees to extend their proboscis to a colorful piece of paper dipped in a 60% sucrose solution (*w/v*). We presented the bee with either a blue or yellow piece of paper dipped in sucrose, allowed it to feed for 5 s, then removed the paper. We repeated this process once every 5 min for 5 iterations. After another 5 min, we presented the bee with the color to which they were trained and the novel color, both dipped in water, and recorded which color the bee first extended its proboscis to. The bees were scored on a binary scale, with a score of 1 indicating that the bee extended its proboscis to the color to which it was trained, or a score of 0 meaning the bee extended its proboscis to the novel color or did not extend its proboscis to either color after 5 min. At the end of the assay, we gave the bees a 30-min resting period under red light. We then flash froze the bees using ethanol mixed with dry ice and placed them in individual, labeled tubes and stored them at −80 °C for future molecular analyses.

### 2.3. Statistical Analyses

All analyses were performed using R v3.6.1. To conduct a power calculation, we used the package ‘pwr.’ In order to determine whether there were any effects of treatment on survival, we recorded mortality daily during the experimental period. We analyzed differences in mortality between the experimental treatments using a Kaplan–Meier survival analysis with the ‘survival’ R package. Using the ‘survdiff’ function, we analyzed significance with a log-rank test including colony of origin as a random variable and microbe treatment as a fixed variable. We graphed the mortality data using the ‘survminer’ package [32]. To analyze differences between treatments in the behavioral assay, we performed binomial logistic regression with generalized linear mixed models (GLMM) using the packages ‘lme4′ and ‘blme’ [33,34]. Due to a limited number of assay tubes, not all bees could be tested on the final day of the experiment. Some bees were assayed on day 7 or day 9 rather than day 8 (the end of the experimental period). To account for this difference, we initially included age and colony of origin as random effects with microbe treatment as a fixed effect in the model. Because we conducted two separate rounds of the experiment, the low sample size first experiment (round 1) and the augmented sample size second experiment (round 2), we included experimental round as a fixed effect in the model as well. We selected our best fitting models by comparing AIC scores. In order to test the validity of our behavioral assay, we performed a chi-squared test on the choice assay to test whether the bees that participated in the assay were trainable, meaning that they chose the color they were trained to more frequently than the novel color. We made all of our graphs using the ‘ggplot2′ package in R. Raw data and R code for analyses can be found in the Appendix A.

## 3. Results

In total, we performed FMPER assays on 131 bees, 66 microbe-depleted and 65 microbe-inoculated. Out of the 131 bees subjected to the assay, 28 did not participate, meaning that even after extending the starvation period, the bees did not respond to the sucrose stimulus. A further 10 bees responded to neither color during the choice assay. As bees that responded to neither color demonstrated a lack of participation in the assay toward the end, these bees were also incorporated into the participation analyses. Bees that did not participate were excluded from the choice analyses.

We found that bees chose the color they were trained to significantly more frequently than bees that chose the novel color (chi-squared test, chisq = 7.8387, df = 1, *p* = 0.0051). We selected the best fitting models as those that contained colony, but not age, as a random effect had higher ∆AIC scores than other models (participation model: ∆AIC = 0.5663, choice model: ∆AIC = 0.1639). There were no significant differences in participation between treatments (binomial GLMM by maximum likelihood, std. error = 0.4111, z = 0.911, *p* = 0.362, Figure 2). Further, participation in the assay did not differ between the two rounds of the experiment (binomial GLMM by maximum likelihood, std. error = 0.7043, z = −0.156, *p* = 0.876). Of the remaining bees that participated, neither treatment (binomial GLMM by maximum likelihood, std. error = 0.4501, z = −0.015, *p* = 0.988, Figure 3) nor experimental round (std. error = 0.5255, z = 0.924, *p* = 0.356) influenced whether bees extended their proboscis to the color they were trained or to the novel color.

There was, however, an effect of treatment on mortality during round 2 (Kaplan–Meier log-rank test, chi-squared = 42.3, df = 9, *p* < 0.0001) of the experiment, but not during round 1 (Kaplan–Meier log-rank test, chi-squared = 4.3, df = 7, *p* = 0.7) (Figure 4A,B). In round 2, the significance in mortality is driven by the microbe-inoculated treatment in two of the four colonies that were used (colonies L4 and L6), with greater mortality in the microbe-inoculated bees. Qualitative microscopy revealed live *C. bombi* cells in several bees from each colony, indicating a potential colony-wide infection in all experimental colonies.

## 4. Discussion

Contrary to our prediction, the gut microbiota did not play a significant role in visual learning and memory in our assays using adult bumble bees. Our negative result contradicts the recent findings in *D. melanogaster* [12]. The differences in our results may stem from several possible explanations. First, if our results are generalizable across studies and bumble bee species, it may be that the gut–brain axis is not conserved across all insects. Alternatively, experimental details may explain the differences in our studies. DeNieu et al. reared truly gnotobiotic flies by feeding them tetracycline-spiked media for multiple generations [12]. We did not quantify the microbes in our experimental bees, and it is possible that our microbe-depleted samples harbored a similar abundance of microbes as our microbe-inoculated bees. Our approach is not without precedent; Zheng and colleagues used the same germ-free methods that were verified in a former study without reverification in a later study, as we have done here [18]. In our previous work, we found that our protocol greatly reduced the diversity and abundance (by 10^4^ times) of microbes in the gut [22], although it is unlikely that we can produce a truly gnotobiotic bee without the use of antibiotics. DeNieu et al. also found that as few as two species of bacteria could rescue the detrimental effects induced by the absence of microbes [12]. It is, therefore, possible that the trace number of microbes in the guts of our microbe-depleted bees is sufficient to prevent detrimental effects on learning and memory. It is also possible that a visual learning assay is not the most appropriate test for this phenomenon. Other studies have suggested that bees learn visual cues more readily with operant conditioning [35]. Behavioral assays using odor-based learning may be more informative. Our results suggest that there is no strong gut–brain axis in *B. impatiens*, but further experiments are required before this hypothesis can be soundly rejected.

*Crithidia bombi* was also present in all of our experimental colonies. As *C. bombi* affects learning and flower handling in bumble bees [26] and our microbe inoculations likely passaged *C. bombi* to our experimental treatment, our results may be confounded by this parasite. The bumble bee gut microbiome, however, lowers the intensity of infection of *C. bombi* [20]. Further experiments examining the gut–brain axis, gut microbes, and gut parasites will help disentangle the web of complex interactions occurring in the bumble bee gut.

Interestingly, there was a significant effect of treatment on mortality in the second experimental round. In this case, there was increased mortality in microbe-inoculated samples when compared to microbe-depleted samples. It is possible that *C. bombi* infection caused higher mortality in the microbe-inoculated bees in the second round of our experiment, but we note that *C. bombi* is virulent only in the presence of other stressors such as starvation [36]. As bees in both rounds of our experiments were potentially exposed to *C. bombi,* offered food ad libitum, and kept at optimal temperature and humidity settings, this may be an unlikely explanation. Alternatively, the increased mortality in round two may be a result of improper handling when dissecting pupae out of cocoons. However, given that mortality differs between treatments, all bees were treated under the same conditions and randomly assigned to treatments, and as the position of bees in the incubator was rotated daily to avoid microclimatic effects in the incubator, this also seems an unlikely explanation. As an explanation for this increased mortality is therefore not readily apparent, further work is needed to understand what might be causing this result.

## 5. Conclusions

We have presented here the first study, to our knowledge, that has examined whether a brain-gut-microbiome axis exists in bees. We performed a lab-based behavioral assay to test whether this axis is present and can influence learning and memory under experimental conditions. In summary, we found that the presence of the bumble bee gut microbiota may not be an important player in visual learning and memory. While this is true in the context of our study, we emphasize that further work in bumble bees is needed to fully answer this question. Mounting evidence in both vertebrates and invertebrates suggests that the gut microbiota can play an important role in cognitive functions. Future studies should examine whether this pattern persists under different experimental conditions and in other invertebrate systems.

## Figures and Tables

**Figure 1 insects-11-00517-f001:**
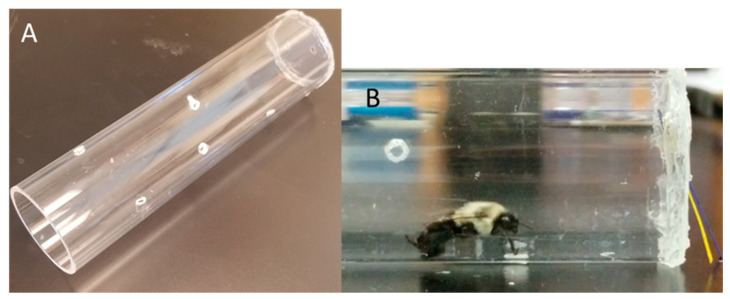
(**A**) The modified tubes used for the behavioral assay. (**B**) An adult bumble bee during the behavioral assay presented with both colors of paper.

**Figure 2 insects-11-00517-f002:**
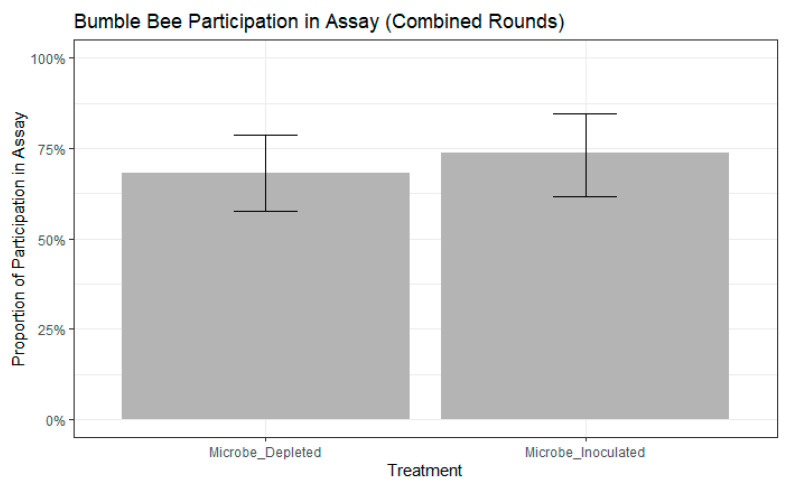
Bumble bee participation in free-moving proboscis extension response (FMPER) assay. No significance between treatments as determined by binomial generalize linear mixed model (GLMM) with maximum likelihood: std. error = 0.4111, z = 0.911, *p* = 0.362. Error bars indicate 95% confidence intervals.

**Figure 3 insects-11-00517-f003:**
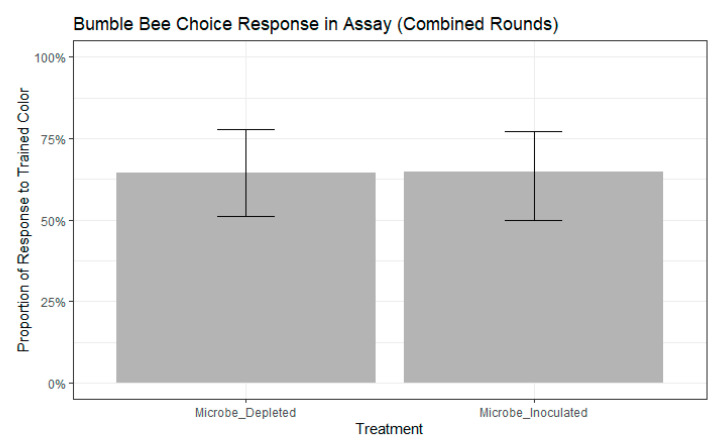
Bumble bee choice response to the trained color, the novel color, or neither color during the FMPER assay. No significance between treatments as determined by binomial GLMM with maximum likelihood: std. error = 0.4501, z = −0.015, p = 0.988). Error bars indicate 95% confidence intervals.

**Figure 4 insects-11-00517-f004:**
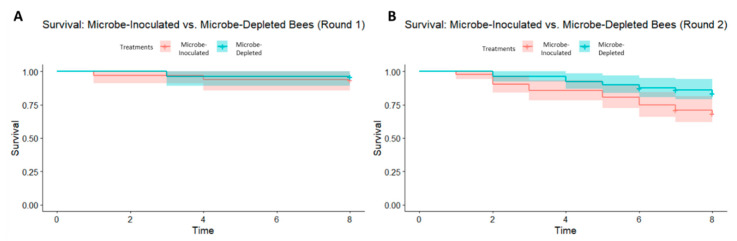
(**A**) Kaplan–Meier curve of bumble bee survival probability from experimental round 1 (Kaplan–Meier log-rank test, chi-squared = 4.3, df = 7, *p* = 0.7) and (**B**) experimental round 2 (Kaplan–Meier log-rank test, chi-squared = 42.3, df = 9, *p* < 0.0001) with 95% confidence intervals.

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
