# Peer review of "The Gut–Brain–Microbiome Axis in Bumble Bees"

_insects, 2020, doi:10.3390/insects11080517_

Round 1

Reviewer 1 Report

The gut-brain-microbiome axis in bumble bees

The authors experimentally tested whether the gut microbiota influences learning and memory in adult bumble bees thus demonstrating gut-brain-microbiome axis in bees.

The paper is well written in clear, concise language.  The figures illustrate the results well. The introduction gives an interesting summary of the functions of gut microbiota in bees as we understand them currently. I would endorse the paper being published as submitted except for one comment below.

Comment: The Bombus sp. used in these experiments is not stated in the methods section and it is not used in the result section. This should be included in the paper. Yes, it is stated that the bees came from Koppert but there may be some readers that do not know what this means (for example, researchers who work on Drosophila). 

Author Response

Please find our responses to these constructive comments below in bold.

Reviewer #1:

Comments and Suggestions for Authors

The authors experimentally tested whether the gut microbiota influences learning and memory in adult bumble bees thus demonstrating gut-brain-microbiome axis in bees.

The paper is well written in clear, concise language.  The figures illustrate the results well. The introduction gives an interesting summary of the functions of gut microbiota in bees as we understand them currently. I would endorse the paper being published as submitted except for one comment below.

Thank you for the positive feedback. We believe our manuscript will be a meaningful contribution to the field.

Comment: The Bombus sp. used in these experiments is not stated in the methods section and it is not used in the result section. This should be included in the paper. Yes, it is stated that the bees came from Koppert but there may be some readers that do not know what this means (for example, researchers who work on Drosophila). 

 Thank you for catching this oversight. We have adjusted the manuscript to include the bumble bee species in the abstract on lines 15-16:

We hypothesized that because of its importance in bee health, the gut microbiota influences learning and memory in adult bumble bees. As an initial test of whether there is a gut-microbiome-brain axis in bumble bees, we reared microbe-inoculated and microbe-depleted bees from commercial Bombus impatiens colonies.”

We have also included the species in our introduction on lines 70-71:
“We reared microbe-depleted and microbe-inoculated bees from commercial Bombus impatiens colonies and subjected them to a visual learning assay to determine whether the absence of a gut microbiota could induce learning deficits.”

And lastly in our methods section on line 76:

“We performed two separate experiments, each with a different set of Bombus impatiens colonies from the same commercial provider, Koppert Biological Systems (Howell, MI).”

Reviewer 2 Report

This study examines whether the presence of an abundant gut microbiome influences a bumble bee’s cognition. The authors raised bees in sterile conditions, and provide half their bees with a microbial inoculum. They then tested whether the presence of gut microbes influenced how often bees went to a trained reward.

While there are elements of this study that seem well designed, and the general manuscript is easy to follow, I do have some concerns regard the methodology. Specifically, I am skeptical as to whether the “low microbiome” bees were truly lower in microbial abundance, and I also am concerned about the efficacy of the training assay. I have outline these concerns as well as others in increased detail below.

Major Comments:

One of my more major concerns is that there is never an assessment of the bees to determine whether the low inocula specimens were truly lower than the full inocula bees. As the authors are likely aware, maintaining sterility can be difficult, and it is therefore very important to demonstrate that the manipulations of the microbiome were successful.

My second major comment is in regard to the study design. Did you analyze whether your training assay worked, regardless of the treatment? It may appear as though you were able to “train” the bees based on Figure 3, but it would be important to examine this statistically. The main reason being that if none of your bees ever showed the “ability to be trained”, then the design of this study would ultimately be flawed, and no conclusions can be made on whether microbiomes influence cognition.

Minor Comments:

Please provide the species of bee you are using.

You mention your sample size in the “first” experiment, but not in the second. Can you include this information for the second experiment? I realize it is eventually provided in the results section, but it seems more appropriate for the methods section.

Additionally, you mention that you include experimental round as a random variable. Does this refer to the first experimental round of 51 bees and then the second round being your “second experiment”? I don’t believe that there is an issue with this method, but I am having a bit of trouble following your terminology. If experimental round refers to your “first” and “second” experimental groups, then that seems appropriate and necessary to include as a random variable.

The authors need to provide binomial confidence intervals as estimates of the variance. They also should provide variance around their Kaplan-Meier curves, which can be done in the survival package in R.

Figure 2 and 3: This figure seems incorrect because each treatment only equals a total of 0.5 of the proportion. The correct way to demonstrate this would be to show the proportion of trained color versus novel color responses per treatment type as 100% of the proportions. The axis would then go to 1.0 and the difference within treatments would change.

Line 7: THE is bolded

Line 12: should be gut microbiota or the gut microbiome

Line 18: I find the phrasing of this last sentence a bit strange. This study isn’t on vertebrate systems, so to say it is not conserved across vertebrates and all invertebrates is a bit misleading. I believe it would make more sense to state:

While these results suggest that the gut microbiome-brain axis is not evident in (insert this species), further studies are needed to determine whether this phenomenon is present in other invertebrate systems.

Line 25: , after study

Line 51-53: Can you be more explicit with what you are trying to say here? It seems like an important point and to make readers look for it in another manuscript makes it difficult to follow.

Line 121: I appreciate the photos for reference.

Author Response

Please find our responses to these constructive comments below in bold.

Reviewer #2:

Comments and Suggestions for Authors
This study examines whether the presence of an abundant gut microbiome influences a bumble bee’s cognition. The authors raised bees in sterile conditions, and provide half their bees with a microbial inoculum. They then tested whether the presence of gut microbes influenced how often bees went to a trained reward.

While there are elements of this study that seem well designed, and the general manuscript is easy to follow, I do have some concerns regard the methodology. Specifically, I am skeptical as to whether the “low microbiome” bees were truly lower in microbial abundance, and I also am concerned about the efficacy of the training assay. I have outline these concerns as well as others in increased detail below.

Major Comments:

One of my more major concerns is that there is never an assessment of the bees to determine whether the low inocula specimens were truly lower than the full inocula bees. As the authors are likely aware, maintaining sterility can be difficult, and it is therefore very important to demonstrate that the manipulations of the microbiome were successful.

This is an excellent point. We agree that it would strengthen our manuscript to provide qPCR data to show that our non-inoculated samples have lower microbial abundance than or microbe-inoculated samples as we have in our previous experiments (Rothman et al. 2019). However, the journal has requested a fast turnaround and due to quarantine restrictions, it is not possible to provide qPCR data at this time. Using a proven protocol without reverification is, however, not without precedent. The Moran lab, for instance, has used the same approach of using the same protocols from previous publications from the lab that were shown to work in the first publication but not the second publication (Zheng et al. 2017, PNAS). Therefore, we are confident that our protocol worked.

For transparency, we have included details about this in our methods on lines (111-113):
“We used germ-free protocols that have previously been demonstrated to decrease the abundance and diversity of the bee gut microbiome by 104 times
[22].”

And in our discussion section on lines (204-211):
We did not quantify the microbes in our experimental bees, and it is possible that our microbe-depleted samples harbored a similar abundance of microbes as our microbe-inoculated bees. Our approach is not without precedent; Zheng and colleagues used the same germ-free methods that were verified in a former study without reverification in a later study, as we have done here [35]. In our previous work we found that our protocol greatly reduced the diversity and abundance (by 104 times) of microbes in the gut [22], although it is unlikely that we can produce a truly gnotobiotic bee without the use of antibiotics

My second major comment is in regard to the study design. Did you analyze whether your training assay worked, regardless of the treatment? It may appear as though you were able to “train” the bees based on Figure 3, but it would be important to examine this statistically. The main reason being that if none of your bees ever showed the “ability to be trained”, then the design of this study would ultimately be flawed, and no conclusions can be made on whether microbiomes influence cognition.

Thank you for raising this concern. We agree that validating the efficacy of our behavioral assay is necessary to draw appropriate conclusions. To address this, we have performed additional analyses on our data. We performed a chi-squared test to compare the trained bees that chose the color they were trained to versus those that chose the novel color. We found that significantly more bees chose the color they were trained to than the novel color (chi-squared test, chisq = 7.8387, df = 1, p = 0.005114). We removed the bees that chose neither color from this analysis as they stopped participating in the assay at that point rather than showing evidence of being trained or not trained. We have removed the same bees that chose neither color from the choice analyses based on the same rationale and incorporated them into our participation analyses. We have re-run the analyses with these adjustments and found that our results do not change. We have updated our figures accordingly.

We have updated the text to include these new analyses in our methods on lines (159-162):
“In order to test the validity of our behavioral assay, we performed a chi-squared test on the choice assay to test whether the bees that participated in the assay were trainable, meaning that they chose the color they were trained to more frequently than the novel color.”

We have also updated our results accordingly on lines (166-172):
“A further 10 bees responded to neither color during the choice assay. As bees that responded to neither color demonstrated a lack of participation in the assay toward the end, these bees were also incorporated into the participation analyses. Bees that did not participate were excluded from the choice analyses.

We found that bees chose the color they were trained to significantly more frequently than bees that chose the novel color (chi-squared test, chisq = 7.8387, df = 1, p = 0.005114).”

Minor Comments:

Please provide the species of bee you are using.
Thank you for catching this oversight. We have adjusted the manuscript to include the bumble bee species in the abstract, introduction, and methods sections as described below.

In our abstract on lines (15-16):

We hypothesized that because of its importance in bee health, the gut microbiota influences learning and memory in adult bumble bees. As an initial test of whether there is a gut-microbiome-brain axis in bumble bees, we reared microbe-inoculated and microbe-depleted bees from commercial Bombus impatiens colonies.”

We have also included the species in our introduction on lines (70-72)
“We reared microbe-depleted and microbe-inoculated bees from commercial Bombus impatiens colonies and subjected them to a visual learning assay to determine whether the absence of a gut microbiota could induce learning deficits.”

And lastly in our methods section on lines (76-77):

“We performed two separate experiments, each with a different set of Bombus impatiens colonies from the same commercial provider, Koppert Biological Systems (Howell, MI).”

You mention your sample size in the “first” experiment, but not in the second. Can you include this information for the second experiment? I realize it is eventually provided in the results section, but it seems more appropriate for the methods section.
Great suggestion. We have adjusted our methods section to include the sample sizes of our second experiment on lines (82-83):

“For the second experiment (round 2) we increased our sample sizes to n = 41 microbe-depleted bees and n = 39 microbe-inoculated bees.”

Additionally, you mention that you include experimental round as a random variable. Does this refer to the first experimental round of 51 bees and then the second round being your “second experiment”? I don’t believe that there is an issue with this method, but I am having a bit of trouble following your terminology. If experimental round refers to your “first” and “second” experimental groups, then that seems appropriate and necessary to include as a random variable.
Thank you for this insight. Yes, we intended this to be interpreted as the first experimental round of 51 bees and the second experiment with an expanded sample size. After researching further, we concluded it was best to include ‘experimental round’ as a fixed effect in our model. As a factor that has only 2 levels, it is more appropriate to use experimental round to determine whether there were differences in bee choice or participation between the experimental rounds, rather than estimating variation across only 2 levels.

We have clarified the language to make this more apparent in our methods on lines (157-159):

“To account for this difference, we included age and colony of origin as random effects with microbe treatment as a fixed effect in the model. Because we conducted two separate rounds of the experiment, the low sample size first experiment (round 1) and the augmented sample size second experiment (round 2), we included experimental round as a fixed effect in the model as well.”

We have also adjusted our results sections to include the new analysis on lines (175-180):
“There were no significant differences in participation between treatments (binomial GLMM by maximum likelihood, std. error = 0.4111, z = 0.911, p = 0.362, Fig. 2). Further, participation in the assay did not differ between the two rounds of the experiment (binomial GLMM by maximum likelihood, std. error = 0.7043, z = -0.156, p = 0.876). Of the remaining bees that participated, neither treatment (binomial GLMM by maximum likelihood, std. error = 0.4501, z = -0.015, p = 0.988, Fig. 3) nor experimental round (std. error = 0.5255, z = 0.924, p = 0.356) influenced whether bees extended their proboscis to the color they were trained or to the novel color.”

Upon further research, we realized that we did not analyze the random effects properly. Because the random effects only contribute to the variance in the model, we have removed the previous results detailed on lines 180-186. that provided a p-value. Initially, we tested the random variables in a separate model as fixed effects to determine whether they influenced the results. We now, instead, use AIC values to compare our models and ensure that we used models with the best fit.

We have adjusted our methods to fit these changes on line 160:
“We selected our best fitting models by comparing AIC scores.”

And in our results on lines 173-176:
“We selected the best fitting models as those that contained colony, but not age, as a random effect had higher ∆AIC scores than other models (participation model: ∆AIC = 0.5663, choice model: ∆AIC = 0.1639).”

The authors need to provide binomial confidence intervals as estimates of the variance. They also should provide variance around their Kaplan-Meier curves, which can be done in the survival package in R.
Great suggestion. We have included confidence intervals as estimates of variance for our models as you have recommended. We have also added confidence intervals to our Kaplan-Meier curves.

We have also adjusted our figure captions to match these changes on lines (357-359):
“Figure 2: Bumble bee participation in FMPER assay. No significance between treatments as determined by binomial GLMM with maximum likelihood: std. error = 0.4111, z = 0.911, p= 0.362. Error bars indicate 95% confidence intervals.”

Lines (360-363):
“Figure 3: Bumble bee choice response to the trained color, the novel color, or neither color during the FMPER assay. No significance between treatments as determined by binomial GLM with maximum likelihood:  std. error= 0.4501, z = -0.015, p= 0.988). Error bars indicate 95% confidence intervals.”

And lines (364-367):
“Figure 4: A) Kaplan-Meier curve of bumble bee survival probability from experimental Round 1 (Kaplan-Meier log-rank-test, chi-squared = 4.3, df = 7, p = 0.7) and B) experimental Round 2 (Kaplan-Meier log-rank-test, chi-squared = 42.3, df = 9, p < 0.0001) with 95% confidence intervals.”

Figure 2 and 3: This figure seems incorrect because each treatment only equals a total of 0.5 of the proportion. The correct way to demonstrate this would be to show the proportion of trained color versus novel color responses per treatment type as 100% of the proportions. The axis would then go to 1.0 and the difference within treatments would change.
Another excellent point. We have adjusted Figures 2 and 3 as you have suggested.

Line 7: THE is bolded
Thank you for catching this formatting error. We have changed this.

Line 12: should be gut microbiota or the gut microbiome
We would like to respectfully point out that the line in question does use the term ‘gut microbiota.’ We are happy to make a change if one is needed here.

Line 12 reports:
We hypothesized that because of its importance in bee health, the gut microbiota influences learning and memory in adult bumble bees.”

Line 18: I find the phrasing of this last sentence a bit strange. This study isn’t on vertebrate systems, so to say it is not conserved across vertebrates and all invertebrates is a bit misleading. I believe it would make more sense to state:

While these results suggest that the gut microbiome-brain axis is not evident in (insert this species), further studies are needed to determine whether this phenomenon is present in other invertebrate systems.
Thank you for this suggestion. We have edited this sentence on lines (19-22):
“While these results suggest that the gut microbiome-brain axis is not evident in Bombus impatiens, future studies with different invertebrate system are needed to further investigate this phenomenon.”

Line 25: , after study
We have fixed this error.

Line 51-53: Can you be more explicit with what you are trying to say here? It seems like an important point and to make readers look for it in another manuscript makes it difficult to follow.
Another great suggestion. We have added to this sentence to extrapolate on this point in lines (51-54):
Honey bees have been suggested to be ideal hosts to test the generality of the microbiome-gut-brain axis across animals because there is mounting evidence that insect behavior may be influenced by mechanisms that are, in part, homologous to those in mammals [7].” 

Line 121: I appreciate the photos for reference.

We thank you for your constructive comments. We are confident that they have helped us to strengthen our manuscript.

Round 2

Reviewer 2 Report

The authors have done a wonderful job of addressing my comments. I thank them for their time in providing a thoughtful response.